# An efficient multi-gram access in a two-step synthesis to soluble, nine-atomic, silylated silicon clusters

**Kevin M. Frankiewicz** [1,2], **Nicole S. Willeit**[1,2], **Viktor Hlukhyy**[1] & **Thomas F. Fässler** [1,2] ✉

Silicon is by far the most important semiconducting material. However, solution-based synthetic approaches for unsaturated silicon-rich molecules require less efficient multi-step syntheses. We report on a straightforward access to soluble, polyhedral $Si_9$ clusters from the binary phase $K_{12}Si_{17}$, which contains both $[Si_4]^{4-}$ and $[Si_9]^{4-}$ clusters. $[Si_4]^{4-}$ ions, characterised by a high charge per atom ratio, behave as strong reducing agents, preventing $[Si_9]^{4-}$ from directed reactions. By the here reported separation of $[Si_4]^{4-}$ by means of fractional crystallisation, $Si_9$ clusters of the precursor phase $K_{12}Si_{17}$ are isolated as monoprotonated $[Si_9H]^{3-}$ ions on a multi-gram scale and further crystallised as their 2.2.2-Cryptate salt. 20 grams of the product can be obtained through this two-step procedure - a new starting point for silicon *Zintl* chemistry, such as the isolation and structural characterisation of a trisilylated $[^{Me}Hyp_3Si_9]^-$ cluster.

With the progressing technologisation of our society and the accompanying miniaturisation of electronic devices, physicists and chemists face new challenges. Silicon stands out as the most important semiconducting material by far. However, traditional manufacturing methods, such as lithography and etching of crystalline silicon (top-down) or Chemical Vapour Deposition (CVD) of volatile silanes for producing nanostructured components, are reaching their limits. Nevertheless, not only manufacturing these materials requires different approaches. Bulk materials and semiconducting materials in the nanometre range significantly differ in their optical and electronic properties (quantum confinement). Molecular precursors could provide an answer to new manufacturing methods and the necessity for targeted investigations of quantum confinement effects in low-dimensional materials (quantum dots, wires and wells). Alongside the targeted synthesis of silicon nanoparticles, defined molecular (silicon) clusters are also considered model systems for studying physical and chemical processes in nanomaterials.

The targeted synthesis of saturated cage oligosilanes[1,2], unsaturated siliconoids[3–5] and *Zintl* clusters[6–8] has been intensively studied in the past decades. In 1970, West et al. achieved the preparation of a cage oligosilane under reductive conditions starting from chlorosilane precursors for the first time[9]. Such *Wurtz*-type couplings or metathesis reactions also provide access to paradigmatic clusters like silaprismanes[10–12], -cubanes[13–16], and -tetrahedranes[17,18] (Fig. 1; II), primarily impressive by their structural beauty. Following West's initial hints towards the synthesis of a permethylated sila-adamantane[19], Marschner et al. established the synthetic pathway to the sila-adamantane derivative I (Fig. 1), representing a molecular fragment of the diamond structure of elemental silicon. *Via* a simple and elegant cascade of silyl abstraction and silylation steps, coupled with subsequent *Lewis* acid mediated isomerisation, this molecule was obtained in a stepwise synthesis from readily available $TMS_4Si$[20]. Later, targeted functionalised sila-diamondoid derivatives were reported as well[21]. Apart from the synthesis of perchlorocyclohexasilane, the disproportionation of the versatile precursor $Si_2Cl_6$ also enables the synthesis of an endohedral, chloride-decorated silafullerane (Fig. 1; III)[22–24]. In addition to the described saturated silicon clusters, Breher, Scheschkewitz, and Lips report on unsaturated so-called siliconoid

[1]Department of Chemistry, TUM School of Natural Sciences, Technical University of Munich (TUM), Lichtenbergstraße 4, D-85748 Garching, Germany.
[2]Wacker Institute of Silicon Chemistry, Technical University of Munich (TUM), Lichtenbergstraße 4, D-85748 Garching, Germany.
✉e-mail: thomas.faessler@lrz.tum.de

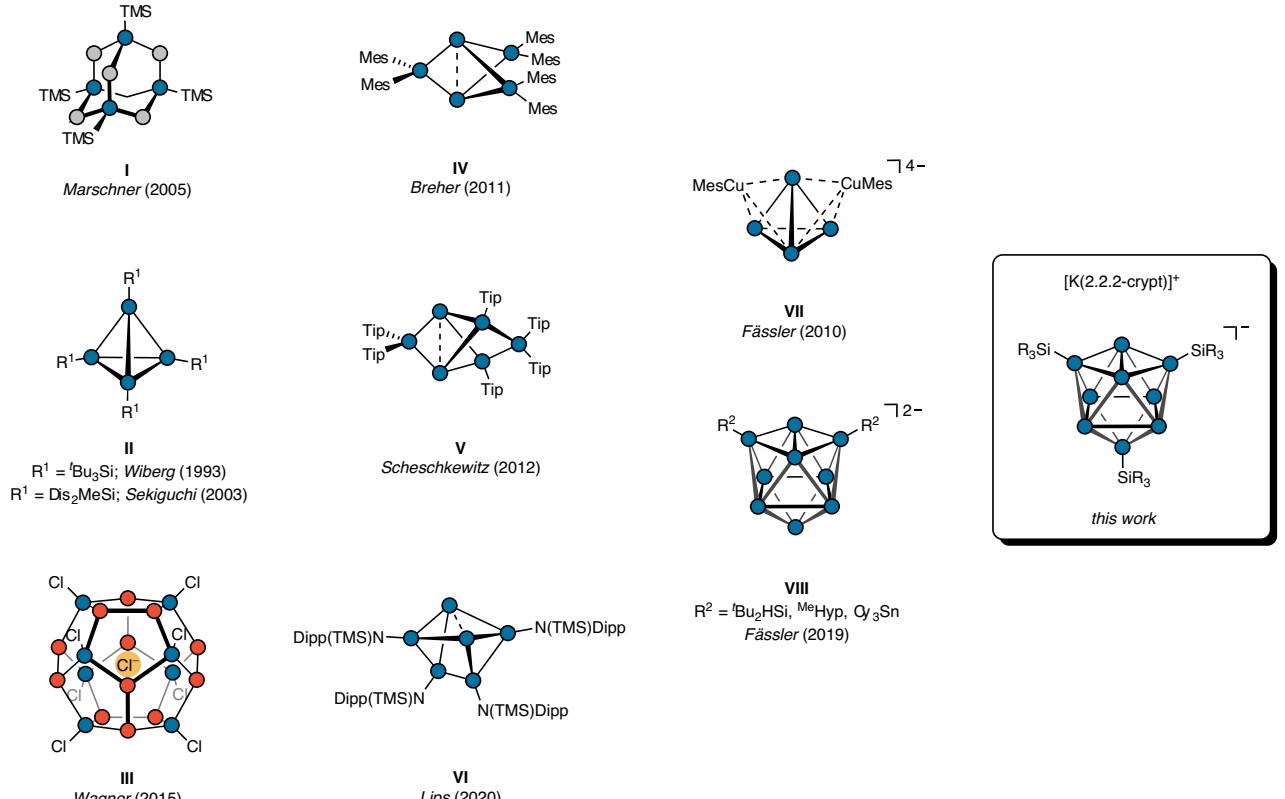

**Fig. 1 | Schematic representation of selected examples of molecular silicon clusters.** Examples **I**–**III** represent saturated, **IV**–**VI** siliconoid and **VII** and **VIII** *Zintl*-type clusters. Silicon is depicted as blue, SiMe₂ units as grey and Si-SiCl₃ units as red circles. TMS Trimethylsilyl, Dis TMS₂HC, Mes 2,4,6-Trimethylphenyl, Tip 2,4,6-Tri-*iso*-propylphenyl, Dipp 2,6-Di-*iso*-propylphenyl, ᴹᵉHyp TMS₃Si, Cy Cyclohexyl.

clusters, a term introduced by Scheschkewitz[11]. Silapropellane **IV** (Fig. 1) is obtained *via* the co-reduction of Mes₂SiCl₂ and Si₂Cl₆, exhibiting a biradicaloid character of the transannular interaction between both bridgeheads[25]. The structurally related, bridged silapropellane **V** (Fig. 1)[26] can be derived from an aromatic dismutational isomer of hexasilabenzene in a thermal or photochemical rearrangement and serves as a starting point for a rich cluster functionalisation and expansion chemistry[27–30]. Furthermore, the suitability of amido ligands[31,32] has been demonstrated in stabilising six-atomic silicon clusters (Fig. 1; **VI**), accessible *via* both the reductive coupling of corresponding bromosilane precursors and the thermal transformation of a zwitterionic tetrasilane[33].

While all discussed routes require multi-step syntheses to build up molecules with multiple Si−Si bonds, an alternative method involves the direct formation of bare silicon clusters in a one-step synthesis by the solid-state reaction of silicon with alkali metals. This approach has yielded considerable success in the case of heavier Ge₉ clusters. For instance, the solid-state phase M₄Ge₉ (M = Na–Cs)[34] can easily be obtained and readily dissolves in various polar solvents. In recent years, a rich chemistry has been established around the nine-atomic [Ge₉]⁴⁻ cluster anion such as (organo)functionalisation[35–50], metalation[43,51–58], and cluster growth[59–65]. In contrast to the heavier tetrel elements (Ge-Pb), [Si₉]⁴⁻ cannot be selectively obtained in a solid-state reaction. Instead, only phases with the composition M₁₂Si₁₇ (equivalent to {(M⁺)₁₂([Si₄]⁴⁻)₂([Si₉]⁴⁻); M = Na–Cs) are accessible, which also contain four-atomic [Si₄]⁴⁻ clusters in addition to the desired nine-atomic clusters[66]. Due to the highly reductive character of these tetrahedral ions, direct reactions of K₁₂Si₁₇ with electrophilic reagents like chlorosilanes are not feasible. As a result, silicon-based *Zintl* cluster chemistry has lagged behind the heavier homologues up to now and is limited to reactions and studies in liquid ammonia. Apart

from bare anions [Si₉]ˣ⁻ (x = 2–4)[67–69] and protonated ions like [Si₄H]³⁻[70] and [Si₉Hₙ]⁽⁴⁻ⁿ⁾⁻ (n = 1, 2)[71–73] there are only six examples of metalated silicon-based *Zintl* ions, such as [PhZnSi₉]³⁻[74], [{Ni(CO)₂}₂(μ-Si₉)₂]⁸⁻[75], [NHCᴰⁱᵖᵖCu(η⁴-Si₉)H]²⁻[76], and [(NHCᵗᴮᵘAu)₆(η²-Si₄)]²⁻[77], as well as [(MesCu)₂Si₄]⁴⁻ (Fig. 1; **VII**)[78] and [NHCᴰⁱᵖᵖCu(η⁴-Si₉)]³⁻[79]. Both the removal of the highly reactive four-atomic silicon clusters and the avoidance of liquid ammonia as a reaction medium are crucial for further developing this kind of chemistry and overcoming synthetic limitations.

In a first step, we recently made the nine-atomic [Si₉]⁴⁻ clusters available for reactions in organic solvents by dissolving K₁₂Si₁₇ in ammonia with 2.2.2-cryptand and subsequent solvent removal. Starting from this so-called activated precursor phase, the extraction of bisprotonated [Si₉H₂]²⁻ clusters in pyridine and the transformation into disubstituted dianions of the form [²R₂Si₉]²⁻ (Fig. 1; **XIII**; ²R = ᴹᵉHyp, ᵗBu₂HSi, Cy₃Sn) in thf was achieved[72,80,81]. However, the four-atomic clusters continue interfering with the respective electrophilic reagents, causing limited functional group tolerance, poor product purity and low yields.

In this work, we report on wet chemical access to a synthetic K₄Si₉ analogue *via* separation of four- and nine-atomic clusters in liquid ammonia through fractional crystallisation. Obtaining such a precursor compound represents a key step in the still largely unexplored chemistry of nine-atomic silicon clusters and allows for the selective synthesis of trisilylated [K(2.2.2-crypt)][(R₃Si)₃Si₉] cluster salts.

## Results

### Separation of [Si₄]⁴⁻ and [Si₉]⁴⁻ in liquid ammonia

Our previous work shows that the solid-state phase K₁₂Si₁₇ can be converted into an activated form, accessible for follow-up reactions like silylation. This conversion is done by dissolving K₁₂Si₁₇ in liquid

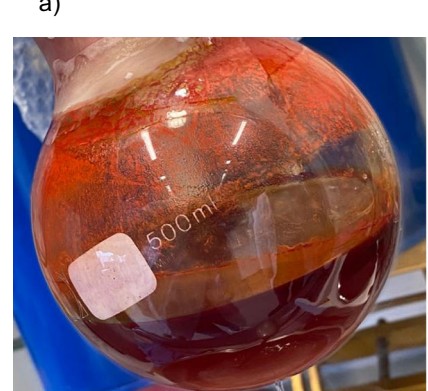
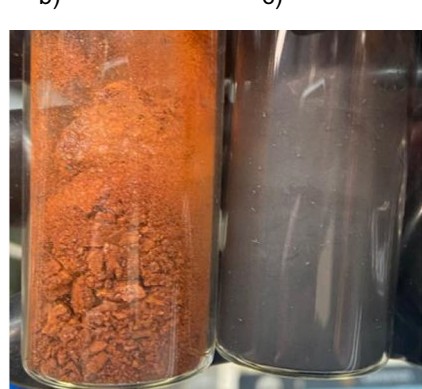

**Fig. 2 | Images of cluster separation. a** Ammonia solution of $K_{12}Si_{17}$ and 2.2.2-cryptand after storage at −40 °C for 12 h; **b** $K_{1-x}[K(2.2.2\text{-crypt})]_{2+x}[Si_9]$ (x = 0.2) obtained from the filtrate after solvent removal; **c** Highly reactive filtration residue after solvent removal.

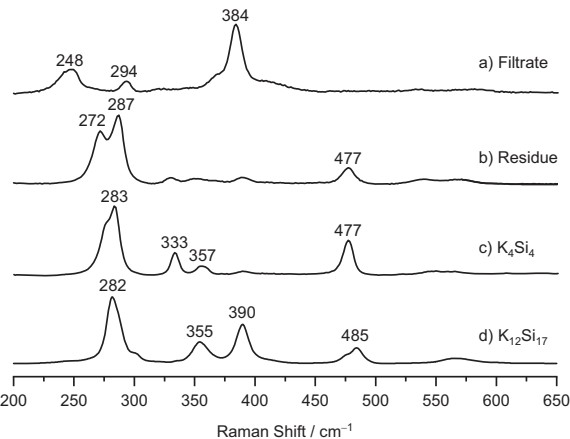

**Fig. 3 | Comparative Raman analysis. a** Filtrate, **b** filtration residue, **c** $K_4Si_4$, and **d** $K_{12}Si_{17}$.

ammonia with 2.2.2-cryptand as a sequestering agent and subsequent solvent removal[80,81]. Nevertheless, interfering four-atomic clusters might still be present in this activated phase.

In order to separate $[Si_4]^{4-}$ and $[Si_9]^{4-}$, we exploit their different solubilities in liquid ammonia. Keeping the ammonia extract of $K_{12}Si_{17}$ for ~12 h at −40 °C, we observe the formation of a bright red solid under a reddish-brown solution (Fig. 2a). After filtration and solvent removal, an orange, coarse solid (Fig. 2b) was isolated from the filtrate. In contrast, the former red filtration residue changed into a grey, finepowder (Fig. 2c). Surprisingly, the Raman measurements of the filtrate and filtration residue after solvent removal (Fig. 3a, b) show a clear separation of the four- and nine-atomic cluster species. For the residue, the most intense resonances at 477 cm⁻¹, 287 cm⁻¹, and 272 cm⁻¹ can be assigned by comparison with the solid-state phase $K_4Si_4$ (Fig. 3c), which contains exclusively four-atomic $[Si_4]^{4-}$ clusters. The slight shift in the resonances is due to the non-identical chemical environment within the crystalline solid and the amorphous filtration residue. In contrast, the Raman spectrum of the filtrate does not show any $Si_4$ band, as can be detected in $K_4Si_4$ and $K_{12}Si_{17}$ (Fig. 3d). The resonances at 294 cm⁻¹ and 384 cm⁻¹ agree with the Raman data already described by *Schnering* for $Cs_4Si_9$, which was obtained by the thermal decomposition of $Cs_4Si_4$[82]. The third resonance at 248 cm⁻¹ indicates a further mode due to the different symmetry with respect to the $C_{4v}$ symmetric $[Si_9]^{4-}$ ion.

Although we can clearly demonstrate that four- and nine-atomic clusters are separated by this procedure, the exact chemical

composition of the dried filtrate cannot be conclusively determined with respect to the number of sequestered cations. We chose the minimum amount of expensive 2.2.2-cryptand, and elemental analysis of the solid shows a composition corresponding to $K_{1-x}[K(2.2.2\text{-crypt})]_{2+x}[Si_9]$ (x = 0.2). The exact 2.2.2-cryptand content in this intermediate after filtration may vary around an ideal composition of $K_1[K(2.2.2\text{-crypt})]_2[Si_9]$, which, however, had no impact on the follow-up chemistry.

Single crystals as orange blocks suitable for SC-XRD were obtained by vapour diffusion of $Et_2O$ into an ammonia solution of the filtrate by adding additional 2.2.2-cryptand to sequester all cations. The structure determination results in the composition of $[K(2.2.2\text{-crypt})]_3[Si_9H]\cdot 8.5\ NH_3$ (**1**). The crystal structure analysis unambiguously shows the formation of a threefold negatively charged cluster. The exceptionally good data quality allows for further structure refinement and the localisation of a hydrogen atom at the cluster from the difference Fourier map and a free refinement of the position of the H atom. The asymmetric unit contains a monoprotonated, threefold negatively charged $[Si_9H]^{3-}$ cluster, three $[K(2.2.2\text{-crypt})]^+$ counter ions and 8.5 equivalents of co-crystallised ammonia. The co-crystallised ammonia primarily occupies the voids between the $[K(2.2.2\text{-crypt})]^+$ units and the cluster. This results in a particular thermal and mechanical sensitivity of the crystals. The refinement shows the presence of an orientationally disordered $C_s$ symmetric $[Si_9H]^{3-}$ cluster (for details, see the Supplementary Discussion). For the main orientation α (Fig. 4), the refinement allows for the localisation of the proton H1A at the Si1A position with a bond length of 1.55(3) Å, which is in the range of typical Si−H bond distances. The cluster framework shows the expected involvement of the substituted Si1A position of the open square plane. Thus, the Si1A-Si2B (2.347(3) Å) and Si1A-Si4 (2.322(2) Å) distances are significantly shortened compared to the Si2B-Si3B (2.580(2) Å) and Si3B-Si4 (2.532(2) Å) distances of the open square plane. The ratio of the square diagonals (Si2B-Si4/Si1A-Si3B) of 1.20 clearly shows the deviation from the ideal $C_{4v}$ symmetry of the parent $[Si_9]^{4-}$ ion[69]. The structural characteristic of shorter Si-Si bonds at cluster atoms with ligands supports the existence of an H atom at Si1A and agrees with findings for the solvate $[K(DB\text{-}18\text{-crown-}6)][K(2.2.2\text{-crypt})]_2[Si_9H]\cdot NH_3$[71,73].

The crystallographic data are supported by mass spectra of acetonitrile (MeCN) solutions of the filtration residue showing $Si_9$ species (Fig. 5), confirming the presence of nine-atomic clusters in the dried filtration product. Additionally, ¹H NMR (Fig. 6) of $K_{1-x}[K(2.2.2\text{-crypt})]_{2+x}[Si_9]$ in DMF-$d_7$ verifies the existence of monoprotonated cluster species. The prominent signal in the high-field region at −1.80 ppm aligns with the spectral range reported for $[Si_9H_2]^{2-}$[72]. The characteristic satellite pattern emerges from scalar coupling to all nine

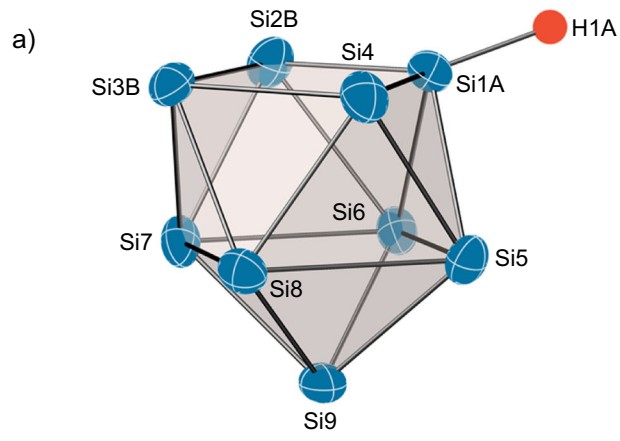

a)

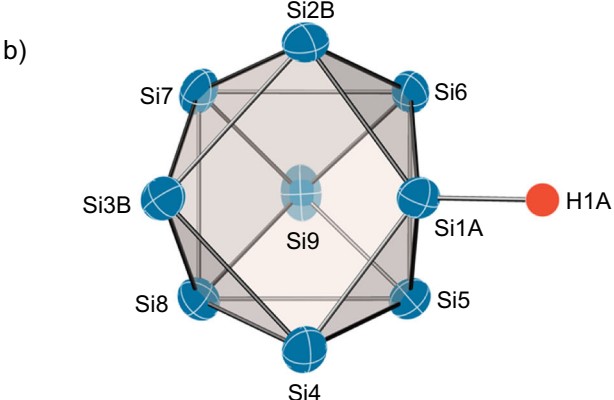

b)

**Fig. 4 | Molecular structure of the anionic cluster moiety [Si₉H]³⁻ (α-orientation) in [K(2.2.2-crypt)]₃[Si₉H]·8.5NH₃ (1). a** Front view; **b** top view. Anisotropic displacement ellipsoids of silicon (blue) are drawn at 50% probability. The hydrogen atom (red) is displayed as a sphere of an arbitrary radius. Silicon and hydrogen atoms of minor disorder components are omitted for clarity. Selected bond length (Å): Si1A-H1A: 1.55(3); Si1A-Si2B: 2.347(3); Si1A-Si3B: 3.121(4); Si1A-Si4: 2.322(2); Si1A-Si5: 2.4244(10); Si1A-Si6: 2.4342(9); Si2B-Si3B: 2.580(2); Si2B-Si4: 3.755(3); Si2B-Si6: 2.5193(12); Si2B-Si7: 2.4655(12); Si3B-Si4: 2.532(2); Si3B-Si7: 2.4553(9); Si3B-Si8: 2.4325(9); Si4-Si5: 2.4905(7); Si4-Si8: 2.4268(7); Si5-Si9: 2.4474(8); Si6-Si9: 2.4326(8); Si7-Si9: 2.4348(8); Si8-Si9: 2.4539(7); Si2B-Si4/Si1A-Si3B: 1.20.

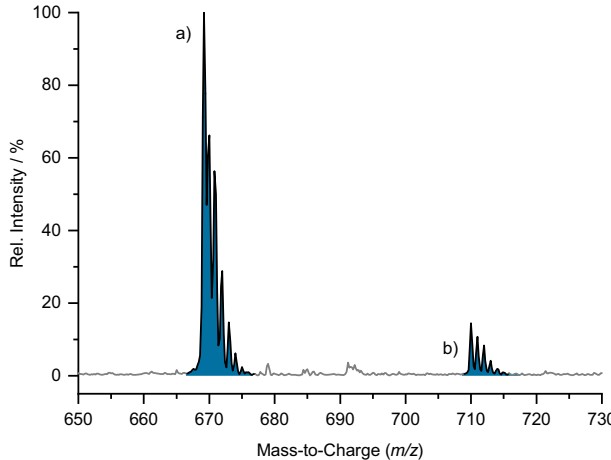

**Fig. 5 | ESI(−) MS spectrum of the dried filtrate in MeCN. a** {[K(2.2.2-crypt)][Si₉] + 2H}⁻ (m/z = 670.37); **b** {[K(2.2.2 crypt)][Si₉] + 2H + mecn}⁻ (m/z = 711.42).

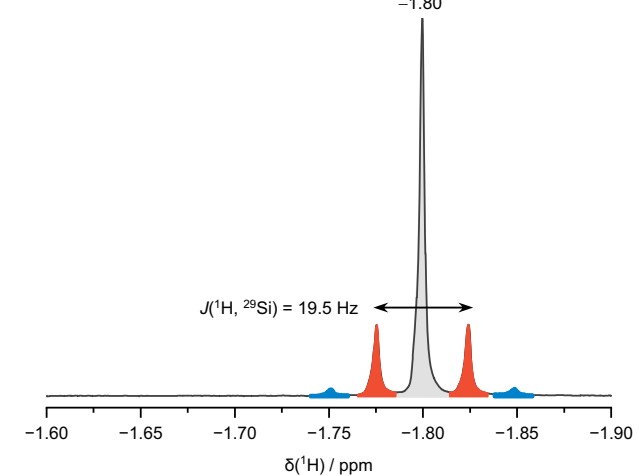

**Fig. 6 | ¹H NMR spectrum of the dried filtrate.** Main resonance of non-coupled isotopologues [²⁸/³⁰Si₉²⁹Si₀H]³⁻ is indicated in grey, doublet splitting of [²⁸/³⁰Si₈²⁹Si₁H]³⁻ in red and visible part of triplet splitting of [²⁸/³⁰Si₇²⁹Si₂H]³⁻ in blue. (400 MHz, DMF-$d_7$, 300 K).

cluster atoms. This pattern displays the superposition of all possible isotopologues caused by the low natural abundance of NMR active ²⁹Si (Natural abundance = 4.7%) in the cluster framework. Consequently, the intense main singlet (indicated in grey) results from all non-NMR active isotopologues ([²⁸/³⁰Si₉H]³⁻). While the first set of satellites (indicated in red) is due to a doublet splitting of the isotopologues [²⁸/³⁰Si₈²⁹Si₁H]³⁻, the second set is due to a triplet splitting (indicated in blue) of the isotopologues [²⁸/³⁰Si₇²⁹Si₂H]³⁻. The satellite signals of higher isotopologues are not detectable due to the low natural abundance of ²⁹Si. A full overview of the statistical intensity distribution for the superposition of all isotopologues is given in the Supplementary Information (Supplementary Fig. 17 and Supplementary Table 7) and in accordance with previous work[72]. The exceptionally small coupling constant of $J$(¹H, ²⁹Si) = 19.5 Hz and the interaction of the proton with all nine silicon atoms of the cluster framework paints the picture of a highly dynamic system at room temperature. At −50 °C, however, a doublet with a significantly increased coupling of 152 Hz is observed (see Supplementary Fig. 18), falling within the typical range of localised ¹$J$(¹H, ²⁹Si) couplings. Thus, proton migration becomes slow on the ¹H NMR timescale at the transition from the high- to the low-temperature limit of proton migration, allowing for direct detection of a localised Si−H unit. This spectroscopic behaviour perfectly aligns with previously reported data in liquid ammonia[71]. Similar ligand migrations have also been described for [Sn₉R³]³⁻ (R³ = H[83], SnCy₃[84]) at room temperature.

In order to clarify the origin of the cluster attached proton, we repeated the whole synthetic protocol of separation of [Si₄]⁴⁻ and [Si₉]⁴⁻ in ND₃ instead of NH₃ as solvent. After this, a signal at −1.62 ppm for [Si₉D]³⁻ can only be detected in the ²H NMR (Supplementary Fig. 19). This shows that the proton of [Si₉H]³⁻ originates from ammonia.

### Reactivity study of [Si₉H]³⁻

The access to isolated Si₉ clusters on a multi-gram scale, in the absence of highly reductive [Si₄]⁴⁻ clusters, provides promising and well-defined conditions for follow-up reactions of nine-atomic silicon clusters. Silylation reactions of K₁₋ₓ[K(2.2.2-crypt)]₂₊ₓ[Si₉] in tetrahydrofuran (thf) lead to the isolation and the structural characterisation of trisilylated cluster salts [K(2.2.2-crypt)][(R₃Si)₃Si₉] (**2a**–**2d**; Fig. 7) in good yields as orange-brown solids. All compounds were characterised by NMR and ESI-MS analyses. Yellow block-shaped single crystals of [K(2.2.2-crypt)][ᴹᵉHyp₃Si₉]·thf were grown from a thf

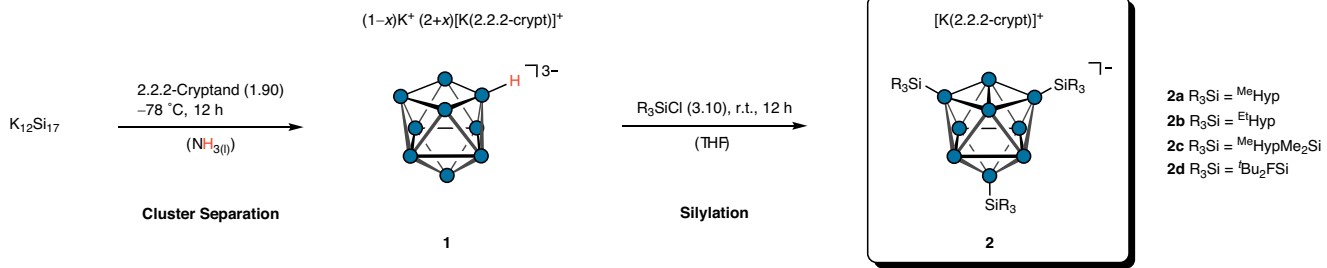

**Fig. 7 | Two-step synthesis of trisilylated silicon clusters 2 *via* protonated cluster species 1.** Silicon is depicted as blue circles. ^RHyp = (R₃Si)₃Si.

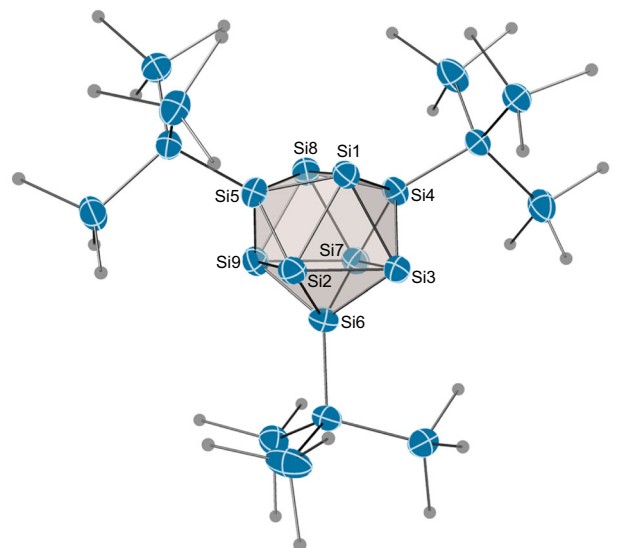

**Fig. 8 | Molecular structure of the anionic cluster moiety [^MeHyp₃Si₉]⁻ in [K(2.2.2-crypt)][^MeHyp₃Si₉]·thf (2a).** Anisotropic displacement ellipsoids of silicon (blue) are drawn at 50% probability. Carbon (grey) is displayed as spheres of an arbitrary radius and hydrogen atoms are omitted for clarity.

## Table 1 | Selected spectroscopic data of 2a–2d

| R₃Si | δ(Si_cap) | δ(Si_cap) | ¹J(²⁹Si_Cap, ²⁹Si_Prism) |
|---|---|---|---|
| ^MeHyp (**2a**) | −175.3 | −360.8 | 40.1 |
| ^EtHyp (**2b**) | −171.3 | −351.8 | 42.7 |
| ^MeHypMe2Si (**2c**) | −146.3 | −347.4 | 24.4 |
| ^tBu₂FSi (**2d**)ᵃ | −177.7 | −357.37 | 23.2 |

ᵃ¹J(¹⁹F, ²⁹Si) = 340.9 Hz; ²J(¹⁹F, ²⁹Si) = 19.4 Hz.
²⁹Si NMR shifts of silylated cluster positions (Si_Cap) and prismatic positions (Si_prism) and homonuclear J coupling (298 K, 79.5 MHz, thf-d₈).

solution at −32 °C over two weeks. The crystal structure of the molecular anion in **2a** is depicted in Fig. 8. **2a** crystallises in the monoclinic space group *P2₁/n* (14) (a = 15.0913(4) Å, b = 24.7859(6) Å, c = 23.9571(6) Å, α = 90°, β = 90.959(2)°, γ = 90°, V = 8959.9(4) Å³) with one trisilylated [^MeHyp₃Si₉]⁻ cluster anion, one disordered [K(2.2.2-crypt)]⁺ unit and one disordered thf molecule in the asymmetric unit (for more details see Supplementary Discussion). Analogously to the homologous germanium cluster[85], the present silicon cluster can also be described as a *D₃ₕ* symmetric threefold capped trigonal prism. The attachment of a further hypersilyl group to the *C₂ᵥ* symmetric dianion [^MeHyp₂Si₉]²⁻[81] leads to a closure of the planar square plane Si1-Si4-Si8-Si5 by shortening of the Si1-Si8 bond from 3.770(7) Å to 3.2565(10) Å. At the same time, the remaining prism edges (Si2-Si9 and Si3-Si7) are elongated by 0.526 Å and 0.430 Å, respectively. The attachment of the third silyl substituent at the Si₉ cluster induces the same geometric changes that have been described for the homologous [^MeHypₙGe₉]^(4-n)− (Supplementary Table 6)[41,85] and [^MeHypₙSn₉]^(4-n)− clusters (n = 2, 3)[86,87]. As expected, the cluster framework undergoes a significant contraction from tin and germanium to silicon. Similar to the homologous germanium cluster [^MeHyp₃Ge₉]⁻[47], the UV-VIS spectra of **2a** and **2d** in thf (Supplementary Figs. 33 and 34) exhibit intense, overlapping signals below 400 nm. The attachment of the electron-withdrawing silyl ligand ^tBu₂FSi in **2d** results in a hypsochromic shift compared to **2a**.

As expected from the crystal structure and analogously to [^MeHyp₃Ge₉]⁻[85], **2a** also behaves *D₃ₕ* symmetric on NMR time scale in

thf-d₈ at room temperature. Hence, the three hypersilyl groups collapse to two ²⁹Si resonances at −130.03 ppm and −8.71 ppm. These signals can be attributed to the *exo*-bonded silicon atoms (*Si*TMS₃) and the TMS groups, respectively. Further, in the high field region at −175.3 ppm, the three capping positions show one signal, while the six equivalent prism positions exhibit a strongly shielded signal at −360.8 ppm. The NMR data are consistent with the data described in our previous studies[81]. The remaining derivatives **2b−d** exhibit similar behaviour, with the cap and prism signals falling within the characteristic range of −160 ppm and −350 ppm. The high quality of the data enables the determination of the ¹J(²⁹Si, ²⁹Si) homonuclear couplings between the cap and prism atoms (Table 1). The coupling constants of the sterically demanding hypersilyl groups (40.1 Hz for **2a** and 42.7 Hz for **2b**) are significantly higher than those of the sterically less demanding silyl groups in **2c** (24.4 Hz) and **2d** (23.2 Hz). These couplings differ from localised Si−Si bonds as in cyclic oligosilanes (¹J(²⁹Si, ²⁹Si) ≈ 50−70 Hz)[88] and may indicate possible dynamic processes in the cluster framework, which have not yet been described in homologous silylated clusters. The access to NMR active cluster frameworks could reveal processes that have remained hidden from us so far.

The presence of strongly reductive [Si₄]⁴⁻ clusters in the solid-state phase K₁₂Si₁₇ limits the directed conversion of nine-atomic [Si₉]⁴⁻ silicon clusters. However, in this work, we have shown that the separation of both cluster species is easily possible on a multi-gram scale in liquid ammonia and provides valuable synthetic access to [Si₉H]³⁻ ions.

[Si₉H]³⁻ shows a pronounced tautomerisation tendency, in which the proton rapidly migrates over the entire nine-atomic cluster framework. In addition, those monoprotonated silicon clusters in the form of the crude product K₁₋ₓ[K(2.2.2-crypt)]₂₊ₓ[Si₉] represent a synthetic equivalent to [Si₉]⁴⁻ ions that are still not accessible in an isolated form *via* a solid-state approach. Thus, the trisilylated cluster salts [K(2.2.2-crypt)][(R₃Si)₃Si₉] (**2a−d**) are obtained in good yields and high purity by direct silylation of K₁₋ₓ[K(2.2.2-crypt)₂₊ₓ[Si₉H] with the corresponding chlorosilanes. The spectroscopic behaviour and the crystallographic characterisation of **2a** prove the strong similarities between silicon- and germanium-based *Zintl* clusters. With the present work, we were able to close a significant gap in the chemistry of group

14 *Zintl* ions. Studies on the further reactivity of the obtained trisilylated monoanions are underway.

# Methods
## General
All reactions and manipulations were performed in oven dried glassware under a purified argon atmosphere using standard *Schlenk* and glove box techniques unless otherwise mentioned. NMR solvents were purchased from Sigma–Aldrich and stored over molecular sieve (3 Å) for at least one day. Dichloromethane, Tetrahydrofuran (THF), and pentane were dried by using a solvent purificator (*MBraun* MB-SPS) and stored over molecular sieve (3 Å). Ammonia was liquified in a dry ice/$^i$PrOH bath and dried over sodium metal for one night prior to use. $ND_3$ was prepared from $D_2O$ and $Mg_3N_2$. Triethylene glycol bis(*p*-toluenesulfonate) was prepared by a modified literature procedure[89].

## Synthesis of K$_{12}$Si$_{17}$
A mixture of potassium (1.49 g, 38.0 mmol, 12.0 eq.) and silicon (1.51 g, 53.9 mmol, 17.0 eq.) was sealed in a tantalum ampule and heated up to 800 °C with a rate of 2 K/min. After 18 h, the reaction mixture was cooled down to room temperature (1 K/min) yielding $K_{12}Si_{17}$ (2.91 g, 97%) as a dark grey microcrystalline solid. The analytical data (Supplementary Fig. 9) agree with the literature[72].

## Synthesis of K$_4$Si$_4$
A mixture of 350 mg potassium (8.95 mmol, 1.00 eq.) and 251 mg silicon (8.95 mmol, 1.00 eq.) was sealed in a tantalum ampule and heated up to 600 °C with a rate of 2 K/min. After ten hours, the reaction mixture was cooled down to room temperature (1 K/min) yielding $K_4Si_4$ (589 mg, 98%) as a dark grey microcrystalline solid. The analytical data (Supplementary Fig. 10) agree with the literature[72].

## Synthesis of triethylene glycol bis(*p*-toluenesulfonate)
45.1 g triethylen glycol (300 mmol, 1.00 eq.) was dissolved in 300 mL $CH_2Cl_2$ under non-inert conditions. After addition of 114 g TsCl (600 mmol, 2.00 eq.), the mixture was cooled to 0 °C and 135 g powdered KOH (2.40 mol, 8.00 eq.) was carefully added in small portions (*Caution: Can cause strong heat evolution*). After stirring for three hours at 0 °C, 300 mL $CH_2Cl_2$ and 600 mL ice-water were added. The organic layer was separated and the aqueous phase was extracted with $CH_2Cl_2$ (3 × 200 mL). The combined organic layers were washed with water (2 × 100 mL), dried over $Na_2SO_4$ and rotary evaporated. Triethylene glycol bis(*p*-toluenesulfonate) (118 g, 258 mmol, 86%) was obtained as a white solid. **$^1$H NMR** (400 MHz, CDCl$_3$, 298 K): δ 7.77 (d, $J = 8.0$ Hz, 4 H), 7.33 (d, $J = 8.0$ Hz, 4 H), 4.12 (t, $J = 4.8$ Hz, 4 H), 3.63 (t, $J = 4.8$ Hz, 4 H), 3.50 (s, 4 H), 2.42 (s, 6 H); **$^{13}$C{$^1$H} NMR** (100 MHz, CDCl$_3$, 298 K): δ 145.0, 133.0, 129.9, 128.0, 70.7, 69.3, 68.8, 21.7. The analytical data agree with the literature[89].

## Synthesis of 2.2.2-cryptand
A mixture of 31.0 g triethylene glycol bis(*p*-toluenesulfonate) (67.0 mmol, 2.00 eq.), 62.1 g of $Na_2CO_3$ (586 mmol, 17.5 eq.) and 4.89 mL 2,2′-(ethylenedioxy)bis(ethylamine) (33.5 mmol, 1.00 eq.) was refluxed in 1000 mL MeCN for five days under non-inert conditions. After cooling to room temperature, the mixture was filtrated and rotary evaporated. The resulting orange oil was redissolved in 375 mL EtOH and 50.0 mL citric acid (1.8 M), heated to 85 °C for three hours and filtrated again. After adjusting the pH of the filtrate to 14 with aqueous tetramethylammonium hydroxide solution, the mixture was rotary evaporated. The resulting residue was redissolved with *Celite* in $CH_2Cl_2$ and rotary evaporated again. After *Soxhlet* extraction with cyclohexane overnight and recrystallisation from $CH_2Cl_2$:$Et_2O$ (1:4), 2.2.2-cryptand was obtained as a white crystalline solid (5.29 g, 14.1 mmol, 42%). Further purification was achieved *via* sublimation (0.1 mbar, 130 °C). The analytical data match with an authentic sample of 2.2.2-cryptand. **$^1$H NMR** (400 MHz, CDCl$_3$, 298 K): δ 2.62 (t, $^3J = 5.56$ Hz, 12H, NC$H_2$CH$_2$), 3.56 (t, $^3J = 5.56$ Hz, 12H, NCH$_2$C$H_2$), 3.66 (s, 12H, CH$_2$). **Elemental Analysis**: (calcd., found for C$_{18}$H$_{36}$N$_2$O$_6$) C (57.42, 57.52), H (9.64, 9.64), N (7.44, 7.44).

## Cluster separation in liquid ammonia
$K_{12}Si_{17}$ (25.0 g, 26.4 mmol, 1.00 eq) and 2.2.2-cryptand (18.9 g, 50.2 mmol, 1.90 eq) were dissolved in 250 mL NH$_{3(l)}$ at −78 °C under inert atmosphere leading to a deep red dispersion. The reaction was stirred for one hour before it was stored at −40 °C for twelve hours. After filtration under continuous cooling with dry ice, the ammonia of the deep red filtrate was evaporated. The resulting red-orange solid was weighted (20.4 g) and used without further purification for silylation experiments. According to the elemental analysis, the solid consist of $K_{1-x}[K(2.2.2\text{-crypt})]_{2+x}[Si_9H]$ ($x = 0.2$). We would like to point out that the number of non-sequestered and sequestered potassium ions may not always have the same ratio and the exact composition of the crude product might slightly vary with respected to the number of sequestered cations. The amount of 2.2.2-cryptand was optimised in order to reach the best separation of Si$_9$ and Si$_4$ clusters also considering using the minimum amount of 2.2.2-cryptand to reduce the costs. We found that the follow-up chemistry of the anion $[Si_9H]^{3-}$ is not influenced by the amount of 2.2.2-cryptand. *Caution: The grey, dried filtration residue reacts explosively with air and protic solvents. Isolation is strongly discouraged. Even Raman measurements conducted within airtight glass capillaries have sometimes resulted in the detonation of these capillaries. Hence, it is strongly recommended to carefully quench the **undried residue** with $^i$PrOH at −78 °C overnight! The solid passivates in $^i$PrOH. After one night, a bright red reactive solid may remain in the flask. Do not quench this solid with water or $^i$PrOH at room temperature under any circumstances! Even small amounts of this residue can react explosively.*

Red orange block-shaped single crystals (20%) of [K(2.2.2-crypt)]$_3$[Si$_9$H]•8.5NH$_3$ (**1**) were obtained by vapour diffusion of Et$_2$O into an ammonia solution of $K_{1-x}[K(2.2.2\text{-crypt})]_{2+x}[Si_9H]$ (1.00 eq.) and cryptand (1.00 eq.) at −40 °C after one week. **$^1$H NMR** (400 MHz, DMF-$d_7$, 298 K): δ −1.80 (s, Si–*H*). **ESI-MS** (negative mode, 3500 V, 300 °C): $m/z$ 670.37 ({[K(2.2.2-crypt)][Si$_9$]+2H}$^-$), 711.42 ({[K(2.2.2-crypt)][Si$_9$]+2H+mecn}$^-$); **Elemental Analysis** (calcd., found for $K_{1-x}[K(2.2.2\text{-crypt})]_{2+x}[Si_9H]$; $x = 0.2$; C$_{39.6}$H$_{80.2}$K$_3$N$_{4.4}$O$_{13.2}$Si$_9$): C (39.70, 39.66), H (6.78, 6.74), N (5.47, 5.14).

## Synthesis of $^{Me}$HypMe$_2$SiH
1.50 g TMS$_4$Si (4.68 mmol, 1.00 eq.) and 551 mg KO$^t$Bu (4.91 mmol, 1.05 eq.) were dissolved in 7.50 mL THF and stirred for 5 h at room temperature. The resulting yellowish solution was slowly added to a solution of 442 mg Me$_2$SiHCl (4.68 mmol, 1.00 eq.) in 5.00 mL THF at −78 °C. After complete addition, the reaction mixture was stirred at room temperature overnight before quenched with sat. aqueous NH$_4$Cl solution. The mixture was extracted with Et$_2$O (3 × 25 mL). The combined organic layers were washed with brine and dried over Na$_2$SO$_4$. After filtration rotary and evaporation of the solvent, $^{Me}$HypMe$_2$SiH (1.32 g, 4.34 mmol, 93%) was obtained as colourless solid. **$^1$H NMR** (400 MHz, CDCl$_3$, 298 K): δ 4.02 (sept, $^3J(^1$H, $^1$H) = 4.4 Hz, dsept, $^1J(^1$H, $^{29}$Si) = 177 Hz, $^3J(^1$H, $^1$H) = 4.4 Hz, 1H, Si–*H*), 0.26 (d, $^3J(^1$H, $^1$H) = 4.4 Hz, 6H, SiMe$_2$), 0.21 (s, 27H, $^{Me}$Hyp); **$^{13}$C{$^1$H} NMR** (101 MHz, CDCl$_3$, 298 K): δ 2.63 ($^{Me}$Hyp), −1.95 (SiMe$_2$); **$^{29}$Si{$^1$H} INEPT** (79.5 MHz, CDCl$_3$, 298 K): δ −9.42 (TMS), −33.51 (SiMe$_2$), −136.59 (TMS$_3$*Si*).

## Synthesis of $^{Me}$HypMe$_2$SiCl
1.00 g $^{Me}$HypMe$_2$SiH (3.26 mmol, 1.00 eq.) and 277 mg TCCA (1.19 mmol, 0.37 eq.) were stirred in 2.00 mL CH$_2$Cl$_2$ overnight under formation of a white suspension. After solvent removal under reduced pressure, the resulting white solid was extracted with pentane (3 × 15 mL). The combined solutions were evaporated under reduced

pressure, giving $^{Me}HypMe_2SiCl$ as colourless solid. **$^1$H NMR** (400 MHz, MeCN-$d_3$, 298 K): δ 0.61 (s, 6H, SiMe$_2$), 0.26 (s, 27H, $^{Me}Hyp$). The analytical data agree with the literature[90].

## General procedure for the synthesis of [K(2.2.2-crypt)][(R$_3$Si)$_3$Si$_9$] (2)

K$_{1-x}$[K(2.2.2-crypt)]$_{2+x}$[Si$_9$H] (x = 0.2) (1.00 eq.) and chlorosilane (3.10 eq.) were dissolved in thf and stirred at room temperature under formation of a red-brown solution. After filtration and removing of the solvent *in vacuo*, the resulting solid was washed with pentane. After drying under reduced pressure, the trisilylated cluster salts [K(2.2.2-crypt)][(R$_3$Si)$_3$Si$_9$] (2) were obtained as orange-brown solids.

## Single-crystal X-ray diffraction (SC-XRD)

Crystal preparation was carried out under a continuous flow of cold nitrogen in perfluorinated ether (Galden® LS 230, Solvay Specialty Polymers Italy SpA). For single-crystal data collection, the crystals were fixed on a glass capillary and positioned in a cold stream (150 K) of dried N$_2$ gas. Single-crystal data collection was performed with a STOE StadiVari diffractometer (Mo *Kα* radiation; λ = 0.71072 Å) equipped with a DECTRIS PILATUS 300 K detector.

The X-Area 1.9 software package (*Stoe*) was used for data reduction and absorption correction[91]. Structures were solved by Direct Methods (SHELXS-2014) and refined by full-matrix least-squares calculations against $F^2$ (SHELXL-2014)[92,93]. The positions of the hydrogen atoms were either refined from the difference Fourier map or calculated and refined using a riding model. Unless otherwise stated, all non-hydrogen atoms were treated with anisotropic displacement parameters. The silicon cluster in compound 1 (CCDC 2338275) shows orientational disorder over three orientations and disorder of non-coordinated ammonia molecules. In compound 2 (CCDC 2232604) the disorder of [K(2.2.2-crypt)]$^+$ and thf has been refined by a split layer refinement. For more details see the Supplementary Information (section Crystallographic Data). The crystal structures have been visualised with CrystalMaker® 11.1.1[94] and Diamond 3.2[95].

## Powder X-ray diffraction (PXRD)

The data were collected at room temperature on a *STOE Stadi P* diffractometer (Ge(111) monochromator, Cu Kα$_1$ radiation, λ = 1.54056 Å) with a DECTRIS MYTHEN 1 K detector in Debye–Scherrer geometry. For the measurements, the samples were sealed in glass capillaries (∅ = 0.3 mm). The raw data were processed with WinX-POW[96]. OriginPro 2023 (OriginLab Corporation) was used for visualisation[97].

## Nuclear magnetic resonance spectroscopy (NMR)

$^1$H, $^2$H, $^{13}$C, $^{19}$F and $^{29}$Si NMR spectra were recorded on a Bruker AVIII Ultrashield 400 and AVIII HD 500 Cryo. The signals of the $^1$H NMR spectra were referenced to the residual proton signal and the $^{13}$C-NMR spectra on the $^{13}$C signal of the deuterated solvent. $^2$H (δ (Me$_4$Si-$d_{12}$) = 0 ppm), $^{19}$F (δ (CFCl$_3$) = 0 ppm), and $^{29}$Si (δ (Me$_4$Si) = 0 ppm) were referenced to external standards. Chemical shift values are given in δ values in parts per million (ppm). The coupling constants *J* are given in Hz. Signal multiplicities are abbreviated as follows: s−singlet, d−doublet, t−triplet, q−quartet, sept−septet, dsept−doublet of septet, b−broad. The spectra were processed and visualised with MestReNova 15.0.0[98] and OriginPro 2023 (OriginLab Corporation)[97].

## Raman spectroscopy

Raman measurements were performed with a Renishaw inVia Reflex Raman System with a CCD Detector (Renishaw 266n10 detector) and a 785 nm laser of 500 mW max. power (Software WiRE 5.3 Renishaw) in sealed glass capillaries (∅ = 0.5 mm)[99]. The spectra were visualised with OriginPro 2023 (OriginLab Corporation)[97].

## Electrospray-ionisation mass spectrometry (ESI-MS)

ESI-MS spectra were measured on an HCT instrument (Bruker Inc). The data were processed with Bruker Compass Data Analysis 4.0 SP 5. The dry gas temperature was adjusted to 573 K and the injection speed to 270 µL/s. Data visualisation of the spectra was carried out with the programs OriginPro 2023 (OriginLab Corporation)[97].

## Ultraviolet-visible spectroscopy (UV-VIS)

UV-VIS spectra were recorded on an Agilent Cary 60 UV-Visible spectrophotometer (Agilent Technologies). The absorption spectra were recorded in 1 mm quartz cuvettes (Hellma Analytics) in thf at room temperature. OriginPro 2023 (OriginLab Corporation) was used for visualisation[97].

## Elemental analysis

Elemental analyses were performed by the microanalytical laboratory at the Catalysis Research Center (CRC) of the Technical University of Munich (TUM). The elements C, H, and N were determined by a combustion analyser (EURO-EA, HEKATech).

## Data availability

All data generated or analysed during this study are available in this published article, its Supplementary Information files or from the corresponding authors on request. The X-ray crystallographic coordinates for the structures reported in this study have been deposited at the Cambridge Crystallographic Data Centre (CCDC), under the deposition numbers 2338275 (1) and 2232604 (2a). These data can be obtained free of charge from The CCDC via www.ccdc.cam.ac.uk/data_request/cif.

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

## Acknowledgements
This work was co-funded by the Wacker Institute of Silicon Chemistry (Wacker Chemie AG) and the Technical University of Munich (TUM). The authors thank Ulrike Ammari and Petra Ankenbauer for the execution of the elemental analyses. They further thank B.Sc. Vivienne Wolde and B.Sc. Thanh N. Trân for their assistance in the synthesis of 2.2.2-cryptand.

## Author contributions
K.M.F. conceived and performed the syntheses of cluster compounds and collected the single-crystal X-ray data of **1** and **2a**, solved and refined the structure of **1** and **2a**, performed the ESI-MS and NMR measurements and prepared samples for further analyses. N.S.W. performed the Raman measurements. V.H. reviewed the structural refinement of **2a**. T.F.F. supervised the work. K.M.F. wrote the paper. K.M.F. and N.S.W. conceived and performed the synthesis of 2.2.2-cryptand. All authors approved the submission of the manuscript.

## Funding

## Competing interests
The authors declare no competing interests.
