## [Transparent Peer Review file · Nature Communications]

An Efficient Multi-Gram Access in a Two-step Synthesis to Soluble Nine-atomic Silylated Silicon Clusters

Corresponding Author: Professor Thomas Fässler

Version 0:

Reviewer comments:

Reviewer #1

(Remarks to the Author)

The manuscript by Fässler and co-workers about "An Efficient Multi-Gram 1 Access in a Two-step Synthesis to Soluble Nine-atomic silylated Silicon Clusters" described significant progress in the synthesis and characterization of substituted Zintl anions.

The manuscript is very well written and all experiments are carried out very precisely and the characterization of the novel compounds was done on a very high level.

Despite the challenges that have been faced with the experimental work to obtain the title compounds on multi-gram scale and the efforts that have been undertaken to characterize the compounds on the documented high standard, this reviewer thinks that the current results are not new and important enough for Nature Communications. I rather suggest to publish in *Angewandte Chemie* or *JACS*.

Here are some further points that should be considered:

- In the abstract line 22, Si₉ cluster is too general. Please specify which Si₉ clusters are relevant here.
- How does the protonation of the charged Si₉ clusters occur? This should be examined in more detail using deuterated solvents?
- The term siliconoid was coined by Scheschkewitz. This should be mentioned in the introduction.
- The recent work from the Klausen lab and the Su lab about molecular silicon clusters should be cited as well.
- Line 215-216: Please specify the significant contraction for charged Sn₉ to Si₉ clusters by giving structural differences in Angström.
- Line 240-241: The dynamic process should be specified and examined in detail by variable temperature NMR-spectroscopy and molecular dynamics simulations.
- The references are not provided consistently. The authors are not cited equally and some titles need further editing.

Reviewer #2

(Remarks to the Author)

This manuscript by Fässler and co-workers describes the large-scale synthesis, separation, and characterization of the protonated nine-atom silicon cluster [Si₉H]³⁻ and its subsequent reactivity to afford functionalized clusters of the type [Si₉(SiR₃)₃]⁻. Even though the structure of [Si₉H]³⁻ has been previously reported with different counter-cations (Korber and Gschwind, *ACIE* 2018, 57, 12956), the novelty of this report resides in the fact that [K(2.2.2-crypt)]₃[Si₉H]·8.5NH₃ can be accessed at scale and in relatively good yields, allowing for interesting follow-up chemistry. The authors show that subsequent derivatization is possible affording tris-silylated clusters [Si₉(SiR₃)₃]⁻. Related germanium clusters have been previously reported by Sevov and Schnepf, while Fässler himself has reported trifunctionalized silicon clusters on at least two separate occasions (*Chem. Eur. J.* 2018, 24, 19171; *Chem. Sci.* 2019, 10, 9130). As far as I call tell, spectroscopic and mass-spec data for related anions such as [Si₉(SiMe₄)₃]⁻, [Si₉(SiHMe₂)₃]⁻ and [Si₉(SnCy₃)₃]⁻ has previously been reported by the authors. There is a lot of overlap between this manuscript and previous reports (both by the authors and others), that makes me question the novelty of the reported work. The big breakthrough here, as far as I can tell, is that bulk access to [K(2.2.2-crypt)]₃[Si₉H]·8.5NH₃ makes a lot of the follow-up chemistry easier, and allows for clean access to [Si₉(SiR₃)₃]⁻ clusters. This manuscript also contains the first structurally authenticated cluster of this type, [K(2.2.2-crypt)]

[Si₉(SiMe₄)₃]-THF, although I doubt that this alone is sufficient grounds for publication in Nature Communications. In summary, I have no doubt that this is a significant methodological breakthrough for the groups working in the field, and that it will lead to advances in the follow-up chemistry of [Si₉R₃]- clusters. That being said, I have reservations about the fact two of the clusters reported in the manuscript, [Si₉H]₃- and [Si₉(SiMe₄)₃]-, have previously been reported in the literature, and that the other three clusters are closely related to the latter.

Technically speaking, the work has been carried out to a high standard. There are one or two questions that need addressing prior to publication.

1) The mass spec data reported for [K(2.2.2-crypt)]₃[Si₉H]-8.5NH₃ shows peaks corresponding to [Si₉]^{+2H+}. How confident are the authors that this indeed arises from [Si₉H]₃- and not [Si₉H₂]₂- (a species they have previously reported)? This seems rather ambiguous.

2) The grey residue reported by the authors is consistent with K₄Si₄. Could this compound (or mixture of compounds) be characterized beyond Raman spectroscopy?

3) A percentage yield for the synthesis of [K(2.2.2-crypt)]₃[Si₉H]-8.5NH₃ should be provided.

Reviewer #3

(Remarks to the Author)

The synthesis of silicon clusters have attracted intensive attention, because silicon is the most important semiconducting material. However, solution-based synthetic approaches for unsaturated silicon-rich molecules require less efficient multi-step syntheses. In this paper, Dr Fässler reported on a wet chemical access to a synthetic K₄Si₉ analogue via separation of four- and nine-atomic clusters in ammonia solution. However, the same group have already made the nine-atomic Si clusters for reactions in organic solvents by dissolving K₁₂Si₁₇ in ammonia with 2.2.2-Cryptand (ref. 77-79). They optimized the reaction conditions in this work. Therefore, I do not think this work is suitable for the publication in Nature Communications that seeks important advances.

1, Elemental analysis for K[K(2.2.2-crypt)]₂[Si₉H] has some gap between calculation data and found data.

2, The UV-vis spectra of the silicon clusters were suggested to collect and discuss.

Version 1:

Reviewer comments:

Reviewer #1

(Remarks to the Author)

The authors addressed all of my questions in a sufficient manner. However as suggested in my previous revision and in agreement with the other reviewers the products of the current manuscript are not new enough to allow for publication in Nature Communications where something with more novelty is expected. I suggest to submit the manuscript to a different journal.

Reviewer #2

(Remarks to the Author)

The authors have made many of the suggested corrections. Of particular interest are the deuterium labelling studies suggested by reviewer #1. These certainly improve the manuscript. The mass-spec data are still ambiguous, and the elemental analysis data for K[K(2.2.2-crypt)]₂[Si₉H] is, according to the authors, contaminated by 2.2.2-crypt, and thus does not provide conclusive evidence of the compositional purity of the sample. The authors have ignored the suggestion of reviewer #3 to collect UV-Vis spectra. On the whole, the manuscript has improved.

Reviewer #3

(Remarks to the Author)

I suggest accepting this revised manuscript.

Version 3:

Reviewer comments:

Reviewer #4

(Remarks to the Author)

"I am only providing comments on the crystallography. The manuscript by Fässler et al. examines the crystal structures of two silicon clusters: [K(2.2.2-crypt)]₃[Si₉H]-8.5 NH₃ (compound 1) and [K(2.2.2-crypt)][MeHyp₃Si₉]-THF (compound 2a). The

data quality for compound 1 appears to be superior to that of compound 2a, with the C-C bond precision value for compound 2a being notably poor at 0.0137.

Compound 1 crystallizes in the triclinic centrosymmetric space group P-1 and contains one molecule of the Si₉H cluster with only 67.1% occupancy for the H atom in the asymmetric unit (AU). In addition to the Si₉H cluster, the AU contains three molecules of [K(2.2.2-crypt)] and 8.5 molecules of ammonia. The three cryptand molecules are well-refined without ambiguity. Three Si atoms show positional disorder over two positions with varying occupancies that sum to unity. The H atom bound to Si1A is located in the difference Fourier map with only 67.1% occupancy. The authors could have improved the structure by separating the two conformers of the Si clusters using the PART commands in ShelX and refining them accordingly. This approach might have also helped locate the other H atom position attached to the Si1B atoms with the remaining 32.9% occupancy.

A total of 8.5 ammonia molecules are present in the AU. While the first four ammonia molecules are well-refined with their H atoms, the remaining five do not have H atoms associated with them. The thermal parameters for N5 to N9 atoms have higher Uani values compared to N1 to N4 atoms, which suggests that refining these atoms with lower occupancies could be beneficial. I believe that the weight loss observed during the TGA experiment could be useful to estimate the total ammonia content in the asymmetric unit (AU). The authors should consider the missing H atoms attached to N-atoms (N5, N6, N7, N8, and N9) for molecular formula calculations. Including these H atoms would provide accurate molecular formula, molecular weight, crystal density, and F(000) values. It is recommended that the authors address this issue.

The crystal structure of compound 2a is poorly refined, with a C-C bond precision value of 0.0137, despite the large crystal size. Additionally, the formula sum is not reported correctly in the CIF, leading to incorrect values for formula weight and crystal density. The authors should avoid being casual in reporting such data. The cryptand moiety is highly disordered along with the THF molecule, and the ADPs clearly indicate positional splitting. The authors should model the disordered cryptand and THF molecules, as this would likely minimize the R1 and wR2 values. Numerous alerts in the checkCIF report for this compound, especially Alerts A, B, and C, could be avoided by refining the structure with consideration for atom splitting."

I recommend that the authors re-refine these structures, make the necessary corrections in the crystal data, and resubmit the manuscript.

Version 4:

Reviewer comments:

Reviewer #4

(Remarks to the Author)

In the revised manuscript, the authors have carried out further structure refinement for both compounds, which has helped to improve the data quality and the presentations in the manuscript. I am okay with the revisions and recommend the paper be accepted for publication.

Dear Reviewers!

We thank all reviewers for their helpful comments.

We are especially grateful to the idea to use deuterated solvents to show the origin of the proton source. However, since we used NH_3 as a solvent, these experiments were a challenge and are the reason for the delay of the submission of our revised version. We are happy to include not the ^2H -NMR experiments which show that the proton indeed originates from the solvent.

Major changes in the manuscript are highlighted with yellow background and included also in the point-to-point answers to the reviewers below.

With best wishes

Thomas Fässler

REVIEWER COMMENTS

Reviewer #1 (Remarks to the Author):

Despite the challenges that have been faced with the experimental work to obtain the title compounds on multi-gram scale and the efforts that have been undertaken to characterize the compounds on the documented high standard, this reviewer thinks that the current results are not new and import enough for Nature Communications. I rather suggest to publish in Angewandte Chemie or JACS.

1. In the abstract line 22, Si_9 cluster is too general. Please specify which Si_9 clusters are relevant here.

We have corrected this ambiguity: “[... Si_9 clusters of the precursor phase $\text{K}_{12}\text{Si}_{17}$...]”

2. How does the protonation of the charged Si_9 clusters occurs? This should be examined into more detail using deuterated solvents?

Thank you for raising this important question. Since we used NH_3 as a solvent these experiments were however a challenge and are the reason for the delay of the submission of our revised version. According to your suggestion, we performed the experiment in ND_3 . And indeed, we can show the presence of $[\text{Si}_9\text{D}]^{3-}$ in the ^2H NMR. (Supplementary Figure 8). We have added this to our manuscript in line 190ff.: “In order to clarify the origin of the cluster attached proton, we repeated the whole synthetic protocol of separation of $[\text{Si}_4]^{4-}$ and $[\text{Si}_9]^{4-}$ in ND_3 instead of NH_3 as solvent. After this, a signal at -1.62 ppm for $[\text{Si}_9\text{D}]^{3-}$ can only be detected in the ^2H NMR (Supplementary Figure 8) This shows that protons of $[\text{Si}_9\text{H}]^{4-}$ originates from ammonia.”

3. The term siliconoid was coined by Scheschkewitz. This should be mentioned in the introduction.

We added in line 60: “[...], a term that has been introduced by Scheschkewitz.¹¹”

4. The recent work from the Klausen lab and the Su lab about molecular silicon clusters should be cited as well.

Thank you for sharing this important work with us. We added the citation to our manuscript in line 56: “Later targeted functionalised sila-diamondoid derivatives were reported as well.²¹” We could not find relevant literature from the Klausen lab.

5. Line 215-216: Please specify the significant contraction for charged Sn_9 to Si_9 clusters by giving structural differences in Angström.

We have included in this manuscript already the structural differences for the directly related systems $[\text{Hyp}_3\text{Ge}_9]^-$ and $[\text{Hyp}_3\text{Si}_9]^-$ which is given in Table 1. A discussion between Sn_9 and Si_9 will most probably correspond to the trend $[\text{Hyp}_3\text{Sn}_9]^-$ and $[\text{Hyp}_3\text{Ge}_9]^-$ and will most probably not add much more insight. However, since this will add several new data to Table 1, we are worried that this will distract from the scope of the article which puts the new synthetic methods to front. Therefore, we like to suggest to not include these structural data here, however, will follow if this is suggested also by the editor.

6. The dynamic process should be specified and examined in detail by variable temperature NMR-spectroscopy and molecular dynamics simulations.

Thank you for your suggestion. We have performed variable temperature ^1H NMR measurements (Supplementary Information, Figure 7) and described the results in line 182ff.: "At $-50\text{ }^\circ\text{C}$ however, a doublet with a significantly increased coupling of 152 Hz is observed (see Supplementary Information, Figure 7), falling within the typical range of localised $^1\text{J}(^1\text{H}, ^{29}\text{Si})$ couplings. Thus, proton migration becomes slow on the ^1H NMR timescale at the transition from the high- to the low-temperature limit of proton migration allowing for direct detection of a localised Si-H unit. This spectroscopic behavior perfectly aligns with previously reported data in liquid ammonia⁷¹

7. The references are not provided consistently. The authors are not cited equally and some titles need further editing.

Thank you for pointing this out. We have corrected the inconsistent citations.

Reviewer #2 (Remarks to the Author):

This manuscript by Fässler and co-workers describes the large-scale synthesis, separation, and characterization of the protonated nine-atom silicon cluster $[\text{Si}_9\text{H}]_3^-$ and its subsequent reactivity to afford functionalized clusters of the type $[\text{Si}_9(\text{SiR}_3)_3]^-$. Even though the structure of $[\text{Si}_9\text{H}]_3^-$ has been previously reported with different counter-cations (Korber and Gschwind, *ACIE* 2018, 57, 12956), the novelty of this report resides in the fact that $[\text{K}(2.2.2\text{-crypt})]_3[\text{Si}_9\text{H}]\cdot 8.5\text{NH}_3$ can be accessed at scale and in relatively good yields, allowing for interesting follow-up chemistry. The authors show that subsequent derivatization is possible affording tris-silylated clusters $[\text{Si}_9(\text{SiR}_3)_3]^-$. Related germanium clusters have been previously reported by Sevov and Schnepf, while Fässler himself has reported trifunctionalized silicon clusters on at least two separate occasions (*Chem. Eur. J.* 2018, 24, 19171; *Chem. Sci.* 2019, 10, 9130). As far as I call tell, spectroscopic and mass-spec data for related anions such as $[\text{Si}_9(\text{SiMe}_4)_3]^-$, $[\text{Si}_9(\text{SiHMe}_2)_3]^-$ and $[\text{Si}_9(\text{SnCy}_3)_3]^-$ has previously been reported by the authors. There is a lot of overlap between this manuscript and previous reports (both by the authors and others), that makes me question the novelty of the reported work. The big breakthrough here, as far as I can tell, is that bulk access to $[\text{K}(2.2.2\text{-crypt})]_3[\text{Si}_9\text{H}]\cdot 8.5\text{NH}_3$ makes a lot of the follow-up chemistry easier, and allows for clean access to $[\text{Si}_9(\text{SiR}_3)_3]^-$ clusters. This manuscript also contains the first structurally authenticated cluster of this type, $[\text{K}(2.2.2\text{-crypt})][\text{Si}_9(\text{SiMe}_4)_3]\cdot \text{THF}$, although I doubt that this alone is sufficient grounds for publication in *Nature Communications*. In summary, I have no doubt that this is a significant methodological breakthrough for the groups working in the field, and that it will lead to advances in the follow-up chemistry of $[\text{Si}_9\text{R}_3]^-$ clusters. That being said, I have reservations about the fact two of the clusters reported in the manuscript, $[\text{Si}_9\text{H}]_3^-$ and $[\text{Si}_9(\text{SiMe}_4)_3]^-$, have

previously been reported in the literature, and that the other three clusters are closely related to the latter.

1. The mass spec data reported for $[\text{K}(2.2.2\text{-crypt})]_3[\text{Si}_9\text{H}]\cdot 8.5\text{NH}_3$ shows peaks corresponding to $[\text{Si}_9] + 2\text{H}^+$. How confident are the authors that this indeed arises from $[\text{Si}_9\text{H}]^{3-}$ and not $[\text{Si}_9\text{H}_2]^{2-}$ (a species they have previously reported)? This seems rather ambiguous.

This question holds significant importance. Indeed, based on the ESI-MS data, it is impossible to differentiate between $[\text{Si}_9\text{H}]^{3-}$ and $[\text{Si}_9\text{H}_2]^{2-}$. On one hand appear only mono-charged species in the mass spec here (doubly-charged species will have a very different isotope pattern), on the other hand may protons always be added to the anion during injection/vaporization in the mass spectrometer. Therefore, a conclusion which protonated species is present in the solid cannot be drawn. However, the available mass data do confirm the presence of protonated Si_9 species, and merely supplement the additional results from SC-XRD. Therefore, we had stated in the manuscript: "The crystallographic data are further supported by mass spectra of acetonitrile (MeCN) solutions of the filtration residue showing protonated Si_9 species". Finally the NMR (^{29}Si), and elemental analysis conclusively show that in this instance, $[\text{Si}_9\text{H}]^{3-}$ rather than $[\text{Si}_9\text{H}_2]^{2-}$ is present.

2. The grey residue reported by the authors is consistent with K_4Si_4 . Could this compound (or mixture of compounds) be characterized beyond Raman spectroscopy?

We like to point out that we do not state that the grey residue is K_4Si_4 but that the Raman spectra identify $[\text{Si}_4]^{4-}$ ions by comparison to Raman data of K_4Si_4 . We noticed in the manuscript already: "The slight shift in the resonances is due to the non-identical chemical environment within the crystalline solid and the amorphous filtration residue." Additional characterisations or conversions are currently not feasible. The grey residue exhibits certain shock and impact sensitivity. Consequently, conducting ^{29}Si MAS NMR measurements is impractical, as attempts to pack the sample have resulted in rotor detonation. Even RAMAN measurements were challenging and resulted in the detonation of one capillary.

3. A percentage yield for the synthesis of $[\text{K}(2.2.2\text{-crypt})]_3[\text{Si}_9\text{H}]\cdot 8.5\text{NH}_3$ should be provided.

We apologize for the lack of detail. We have added the yield (20%).

Reviewer #3 (Remarks to the Author):

The synthesis of silicon clusters have attracted intensively attention, because silicon is the most important semiconducting material. However, solution-based synthetic approaches for unsaturated silicon-rich molecules require less efficient multi-step syntheses. In this paper, Dr Fässler reported on a wet chemical access to a synthetic K_4Si_9 analogue via separation of four- and nine-atomic clusters in ammonia solution. However, the same group have already made the nine-atomic Si clusters for reactions in organic solvents by dissolving $\text{K}_{12}\text{Si}_{17}$ in ammonia with 2.2.2-Cryptand (ref. 77-79). They optimized the reaction conditions in this work. Therefore, I do not think this work is suitable for the publication in Nature Communications that seeks important advances.

1. Elemental analysis for $\text{K}[\text{K}(2.2.2\text{-crypt})]_2[\text{Si}_9\text{H}]$ has some gap between calculation data and found data.

The slight deviation observed between the collected data and the calculated values can be attributed to a slight excess of 2.2.2-Cryptand (approx. 0.2 eq) in the filtration product that results in slightly higher C, H and N values.

2. The UV-vis spectra of the silicon clusters were suggested to collect and discuss.

Previous UV/VIS studies on homologous germanium clusters (doi.org/10.1002/zaac.201800293 and doi.org/10.1002/ange.202304088) have not revealed any distinctive absorption behavior that could be used to discuss the electronic properties of these clusters in detail. Assigning transitions is non-trivial due to the complexity of these systems, consequently such results are difficult or even impossible to interpret.

REVIEWER COMMENTS

Reviewer #1 (Remarks to the Author):

The authors addressed all of my questions in a sufficient manner. However as suggested in my previous revision and in agreement with the other reviewers the products of the current manuscript are not new enough to allow for publication in Nature Communications were something with more novelty is expected. I suggest to submit the manuscript to a different journal.

Answer:

We are firmly convinced that the isolation of $[\text{Si}_9\text{H}]^{3-}$ on a synthetic scale with emphasis on the separation from silicon four-atom clusters is a game changer. In contrast to germanium nine-atom clusters which are accessible in solution through a direct synthesis, a similar synthetic protocol did not exist for nine-atom silicon clusters. As a result, over the last decades numerous fascinating reports on the chemistry of Ge clusters have been published, but comparable little has been reported for silicon clusters.

We report on the separation of the desired nine-atom clusters from the four-atom clusters which serves as a foundational step upon opening a systematic investigation of nine-atom silicon clusters, which will trigger numerous further investigations and discoveries.

The importance of this approach becomes particularly evident when aiming to effectively transfer the multi-step follow-up chemistry of homologous nine-atomic germanium Zintl clusters to silicon. Consequently, we believe that our results are not only novel but also highly significant in this field.

Reviewer #2 (Remarks to the Author):

1. The authors have made many of the suggested corrections. Of particular interest are the deuterium labelling studies suggested by reviewer #1. These certainly improve the manuscript. The mass-spec data are still ambiguous, and the elemental analysis data for $\text{K}[\text{K}(\text{2.2.2-crypt})]_2[\text{Si}_9\text{H}]$ is, according to the authors, contaminated by 2.2.2-crypt, and thus does not provide conclusive evidence of the compositional purity of the sample.

Answer:

Thank you for your appreciation of the deuterium labelling studies.

Thank you for giving us once more the opportunity to clarify the important points of the mass spectra. The mass spectrometric analysis of the filtrate primarily provides evidence for the integrity of the nine-atomic cluster framework and clearly shows that intact Si_9 clusters are obtained after filtration. It is important to note that we do not claim at no point in our manuscript that the detection of protonated cluster species in the gas phase is taken as a proof of a specific protonation state ($[\text{Si}_9\text{H}]^{3-}$ or $[\text{Si}_9\text{H}_2]^{2-}$). Proton transfers can always occur during evaporation of the solution in the mass spectrometer. Therefore, this would be an overinterpretation of the mass spectrometric data and is particularly inadmissible. Again, we do not use the mass spectrometric data for the distinction between mono- or bis-protonated clusters, but the data merely confirm the presence of nine-atomic silicon clusters in the filtrate. Only our extensive and unequivocal NMR spectroscopic and SC-XRD studies show that the silicon clusters can be attributed to monoprotonated $[\text{Si}_9\text{H}]^{3-}$ ions rather than $[\text{Si}_9\text{H}_2]^{2-}$ ions in the filtrate. Thus, we can on one hand refine the single proton directly at the cluster framework due to the good data quality, and on the other hand determine the charge state of -3 based on the number of $[\text{K}(\text{2.2.2-crypt})]^+$ ions in the asymmetric unit. Consequently, the mass spectrometric data are just one part of the overall analysis. Any possible ambiguities of this individual analysis are unequivocally excluded by the entirety of all investigations.

The elemental analysis of the unpurified filtrate shows a remarkably high agreement with $\text{K}[\text{K}(\text{2.2.2-crypt})]_2[\text{Si}_9\text{H}]$. We want to emphasise that this is the raw product directly after filtration in liquid ammonia of highly reactive species. Nevertheless, we find the clusters form salts of partially sequestered potassium ions which had been written in a general form as $\text{K}[\text{K}(\text{2.2.2-crypt})]_2[\text{Si}_9\text{H}]$. In a non-crystalline

materials the exact number of crypt molecules can vary. In the present case the elemental analysis would match exactly with $K_{1-x}[K(2.2.2\text{-crypt})]_{2+x}[\text{Si}_9\text{H}]$ for $x = 0.2$. We like to point out however, that the goal of our work is to provide a simple and reliable access to a starting material containing solely silicon nine-atoms clusters, which was impossible until now. We believe it is important to consider the filtrate as an intermediate step from the solid phase to silylated, pure cluster species. Given the reproducibility of our syntheses, the high purities and yields of our silylated target products (**2**), and the remarkably high correspondence of the elemental analysis of the filtrate, we deem the purity of this intermediate product to be sufficiently demonstrated. The fact that we cannot crystallise the filtrate without adding an additional equivalent of 2.2.2-cryptand to form $[K(2.2.2\text{-crypt})]_3[\text{Si}_9\text{H}]$ ($x = 1$) further suggests that the filtrate can be described as $K_{1-x}[K(2.2.2\text{-crypt})]_{2+x}[\text{Si}_9\text{H}]$. The lower amount of crypt for the synthetic protocol however was used to reduce costs. We added in the revised version: "The amount of encapsulated K ions might vary as indicated by the elemental analysis which results in $x = 0.2$."

2. The authors have ignored the suggestion of review #3 to collect UV-Vis spectra.

Answer: Referee #2 takes up here the argument by referee #3. We replied that we do not expect in comparison the Ge₉ clusters *any distinctive absorption behavior that could be used to discuss the electronic properties of these clusters in detail*. In order to fulfill this comment, we recorded UV-VIS spectra in THF for compounds **2a** (Supplementary Figure 22) and **2d** (Supplementary Figure 23). There is a similarity to known spectra of Ge₉ clusters (for a comprehensive collection of spectra see: *Z. Anorg. Allg. Chem.* **2018**, 644, 1337–1343). In both cases, the absorption behavior of the cluster species is complex, showing most probably an overlap of various transitions in the UV range. We included the following section: "Similar to the homologous germanium cluster $[\text{M}^e\text{Hyp}_3\text{Ge}_9]^-$,⁴⁷ the UV-VIS spectra of **2a** and **2d** in THF (Supplementary Figure 22 and 23) exhibit intense, overlapping signals below 400 nm. The attachment of the electron-withdrawing silyl ligand ^tBu₂FSi in **2d** results in a hypsochromic shift compared to **2a**."

3. On the whole, the manuscript has improved.

Answer: We like to thank for this comment.

Reviewer #3 (Remarks to the Author):

I suggest accepting this revised manuscript.

Answer: Thank you very much.

We like to thank the reviewers for their valuable comments and are happy to point some more details which will help to increase the quality of the paper.

REVIEWER COMMENTS

Reviewer #1 (Remarks to the Author):

The authors addressed all of my questions in a sufficient manner. However as suggested in my previous revision and in agreement with the other reviewers the products of the current manuscript are not new enough to allow for publication in Nature Communications were something with more novelty is expected. I suggest to submit the manuscript to a different journal.

Answer:

We are firmly convinced that the isolation of $[\text{Si}_9\text{H}]^{3-}$ on a synthetic scale with emphasis on the separation from silicon four-atom clusters is a game changer. In contrast to germanium nine-atom clusters which are accessible in solution through a direct synthesis, a similar synthetic protocol did not exist for nine-atom silicon clusters. As a result, over the last decades numerous fascinating reports on the chemistry of Ge clusters have been published, but comparable little has been reported for silicon clusters.

We report on the separation of the desired nine-atom clusters from the four-atom clusters which serves as a foundational step upon opening a systematic investigation of nine-atom silicon clusters, which will trigger numerous further investigations and discoveries.

The importance of this approach becomes particularly evident when aiming to effectively transfer the multi-step follow-up chemistry of homologous nine-atomic germanium Zintl clusters to silicon. Consequently, we believe that our results are not only novel but also highly significant in this field.

In order to point this out more we included “*via separation of four- and nine-atomic clusters in ammonia solution by means of fractional crystallization*” in the abstract and in the introduction.

Reviewer #2 (Remarks to the Author):

1. The authors have made many of the suggested corrections. Of particular interest are the deuterium labelling studies suggested by reviewer #1. These certainly improve the manuscript. The mass-spec data are still ambiguous, and the elemental analysis data for $\text{K}[\text{K}(2.2.2\text{-crypt})]_2[\text{Si}_9\text{H}]$ is, according to the authors, contaminated by 2.2.2-crypt, and thus does not provide conclusive evidence of the compositional purity of the sample.

Answer:

Thank you for giving us once more the opportunity to clarify these important points. The mass spectrometric analysis of the filtrate primarily provides evidence for the integrity of the nine-atomic cluster framework and clearly shows that intact Si_9 clusters are obtained after filtration. It is important to note that we do not claim at no point in our manuscript that the detection of protonated cluster species in the gas phase is taken as a proof of a specific protonation state ($[\text{Si}_9\text{H}]^{3-}$). Proton transfers can always occur during evaporation of the solution in the mass spectrometer. Therefore, this would be an overinterpretation of the mass spectrometric data and is particularly inadmissible. Again, we do not use the mass spectrometric data for the distinction between mono- or bis-protonated clusters, but the data merely confirm the presence of nine-atomic silicon clusters in the filtrate. However, our extensive and unequivocal NMR spectroscopic and SC-XRD studies show that the silicon clusters can be attributed to monoprotonated $[\text{Si}_9\text{H}]^{3-}$ ions. In order to point this out more clear we rephrased it to “The crystallographic data are supported by mass spectra of acetonitrile (MeCN) solutions of the filtration residue show Si_9 species (**Error! Reference source not found.**) and confirm the presence of nine-atomic clusters in the dried product of the filtrate.”

The product is not contaminated by 2.2.2-crypt, but the number of non-sequestered and sequestered K ions is not exactly 1:2. We completely agree that the exact composition of the crude product might slightly vary with respect to the number of sequestered cations, which however has no influence on the anions. Based on the elemental analysis we find the composition $K_{1-x}[K(2.2.2\text{-crypt})]_{2+x}[\text{Si}_9\text{H}]$ for $x = 0.2$. In order to obtain a completely pure substance by crystallization, namely $[K(2.2.2\text{-crypt})]_3[\text{Si}_9\text{H}]$ 8.5 NH_3 more 2.2.2-crypt has to be added. We like to point out however, that the goal of our work is to provide a simple and reliable access to a starting material containing solely silicon nine-atoms clusters, which was impossible until now. We also minimized the amount of expensive 2.2.2-crypt. Given the high reproducibility of our syntheses, the high purities and yields of our silylated target products (**2**), and the remarkably high correspondence of the elemental analysis of the filtrate, we are convinced that $K_{1-x}[K(2.2.2\text{-crypt})]_{2+x}[\text{Si}_9\text{H}]$ is the valid description of the raw material.

We rephrased this section in the manuscript: "Although we can clearly demonstrate that four- and nine-atom clusters are separated by this procedure, the exact chemical composition of the dried filtrate cannot be conclusively determined with respect to the number of sequestered cations. We chose the minimum amount of expensive 2.2.2-crypt and elemental analysis of the solid shows a composition corresponding to $K_{1-x}[K(2.2.2\text{-crypt})]_{2+x}[\text{Si}_9]$ with $x = 0.2$. The exact 2.2.2-crypt content in this intermediate after filtration may vary around an ideal composition of $K_1[K(2.2.2\text{-crypt})]_2[\text{Si}_9]$, which however we found, had no impact on the follow-up chemistry. Single crystals as orange blocks suitable for SC-XRD were obtained by vapour diffusion of Et_2O into an ammonia solution of the filtrate by adding additional 2.2.2-crypt to sequester all cations."

We added in the Supporting Information: "According to the elemental analysis the solid consist of $K_{1-x}[K(2.2.2\text{-crypt})]_{2+x}[\text{Si}_9\text{H}]$ with $x = 0.2$. We like to point out that the number of non-sequestered and sequestered K ions may not always have the same ratio and therefore the exact composition of the crude product might slightly vary with respect to the number of sequestered cations. The amount of 2.2.2-Cryptand was optimized in order to reach the best separation of Si_9 and Si_4 clusters also considering using the minimum amount of 2.2.2-Cryptand to reduce the costs. We found that for follow-up chemistry of the anion was not influenced by the amount of 2.2.2-Cryptand. In order to obtain $[K(2.2.2\text{-crypt})]_3[\text{Si}_9\text{H}]$ 8.5 NH_3 more 2.2.2-crypt has to be added. "

2. The authors have ignored the suggestion of review #3 to collect UV-Vis spectra.

Answer: Referee #2 takes up here the argument by referee #3. We replied that we do not expect in comparison the Ge_9 clusters *any distinctive absorption behavior that could be used to discuss the electronic properties of these clusters in detail*. In order to fulfill this comment, we recorded UV-VIS spectra in THF for compounds **2a** (Supplementary Figure 22) and **2d** (Supplementary Figure 23). There is a similarity to known spectra of Ge_9 clusters (for a comprehensive collection of spectra see: *Z. Anorg. Allg. Chem.* **2018**, 644, 1337–1343). In both cases, the absorption behavior of the cluster species is complex, showing most probably an overlap of various transitions in the UV range. We included the following section: "Similar to the homologous germanium cluster $[\text{MeHyp}_3\text{Ge}_9]^-$,⁴⁷ the UV-VIS spectra of **2a** and **2d** in THF (Supplementary Figure 22 and 23) exhibit intense, overlapping signals below 400 nm. The attachment of the electron-withdrawing silyl ligand tBu_2FSi in **2d** results in a hypsochromic shift compared to **2a**."

3. On the whole, the manuscript has improved.

Answer: We like to thank for this comment.

Reviewer #3 (Remarks to the Author):

I suggest accepting this revised manuscript.

Answer: Thank you very much

We like to thank the reviewer for his valuable comments.

REVIEWER COMMENTS

Reviewer #4 (Remarks to the Author):

I am only providing comments on the crystallography. The manuscript by Fässler et al. examines the crystal structures of two silicon clusters: $[\text{K}(2.2.2\text{-crypt})]_3[\text{Si}_9\text{H}] \cdot 8.5 \text{ NH}_3$ (compound 1) and $[\text{K}(2.2.2\text{-crypt})][\text{MeHyp}_3\text{Si}_9] \cdot \text{THF}$ (compound 2a). The data quality for compound 1 appears to be superior to that of compound 2a, with the C-C bond precision value for compound 2a being notably poor at 0.0137.

1. Compound 1 crystallizes in the triclinic centrosymmetric space group P-1 and contains one molecule of the Si_9H cluster with only 67.1% occupancy for the H atom in the asymmetric unit (AU). In addition to the Si_9H cluster, the AU contains three molecules of $[\text{K}(2.2.2\text{-crypt})]$ and 8.5 molecules of ammonia. The three cryptand molecules are well-refined without ambiguity. Three Si atoms show positional disorder over two positions with varying occupancies that sum to unity. The H atom bound to Si1A is located in the difference Fourier map with only 67.1% occupancy. The authors could have improved the structure by separating the two conformers of the Si clusters using the PART commands in ShelX and refining them accordingly.

Answer:

Thank you for this suggestion. We have classified the Si-H units of the three cluster conformers into PART1-3, making the CIF file more structured. However as in general, the division into parts does not affect the refinement process. We had already accurately described the disorder of the cluster by selecting appropriate restraints. Due to the low scattering power of hydrogen and the small fraction in the unit cell, the hydrogen atoms H2A and H3A of the minor conformers beta and gamma (see the picture below) cannot be refined directly. Therefore, we have now fixed H2A and H3A based on the Si1A-H1A unit of the main conformer alpha.

2. A total of 8.5 ammonia molecules are present in the AU. While the first four ammonia molecules are well-refined with their H atoms, the remaining five do not have H atoms associated with them. The thermal parameters for N5 to N9 atoms have higher Uani values compared to N1 to N4 atoms, which suggests that refining these atoms with lower occupancies could be beneficial.

Answer:

We have added more details about the refinement in the Supporting Information: “The occupancies of all nitrogen atoms were initially refined freely. The refinement revealed that all nitrogen positions from N1 to N8 have an occupancy of 1. In contrast, N9 shows an occupancy of 50% due to its location near the inversion center. The elevated anisotropic displacement parameters of some nitrogen atoms is due to a certain degree of disorder of the respective ammonia molecules. Non-coordinated ammonia, similar to water, often exhibits significant disorder. Due to this disorder, we can only refine the hydrogen atoms of ammonia molecules that are coordinated via hydrogen bonds and are, therefore, well-localised. For N3 we were able to refine a disorder over two positions N3A (71(3)%) and N3B (29(3)%), respectively.”

3. I believe that the weight loss observed during the TGA experiment could be useful to estimate the total ammonia content in the asymmetric unit (AU).

Answer:

TGA could indeed be useful for determining the ammonia content. However, compounds crystallised from liquid ammonia are generally unsuitable for this purpose, as they tend to release ammonia in unknown and variable quantities when heated or subjected to a vacuum. Consequently, sample preparation is not possible under these conditions.

4. The authors should consider the missing H atoms attached to N-atoms (N5, N6, N7, N8, and N9) for molecular formula calculations. Including these H atoms would provide accurate

molecular formula, molecular weight, crystal density, and F(000) values. It is recommended that the authors address this issue.

Answer:

Now we have included the lacking H atoms (which should be attached to N-atoms) for molecular formula calculations.

5. The crystal structure of compound 2a is poorly refined, with a C-C bond precision value of 0.0137, despite the large crystal size. Additionally, the formula sum is not reported correctly in the CIF, leading to incorrect values for formula weight and crystal density. The authors should avoid being casual in reporting such data. The cryptand moiety is highly disordered along with the THF molecule, and the ADPs clearly indicate positional splitting. The authors should model the disordered cryptand and THF molecules, as this would likely minimize the R1 and wR2 values. Numerous alerts in the checkCIF report for this compound, especially Alerts A, B, and C, could be avoided by refining the structure with consideration for atom splitting.

Answer:

Thank you for your advice. We have now fully described all the disorders involving cryptand and THF. See the following figures and the Supporting Information.

The disorder model has greatly improved the refinement and minimised both R1 and R2 values. As a result, there are no longer any A-alerts, and only one B-alert remains: 'Missing # of FCF Reflection(s) Below Theta(Min).' This is due to the large unit cell of compound 2a, which causes the low-angle reflections below 2.608 degrees to be covered by the beamstop. The formula sum is now reported correctly.